# An isotropic zero thermal expansion alloy with super-high toughness

Chengyi Yu [1], Kun Lin [1], Qinghua Zhang [2], Huihui Zhu[1], Ke An [3], Yan Chen [3], Dunji Yu[3], Tianyi Li [4], Xiaoqian Fu[5], Qian Yu[5], Li You[1], Xiaojun Kuang [6], Yili Cao[1], Qiang Li[1], Jinxia Deng[1] & Xianran Xing [1] ✉

Zero thermal expansion (ZTE) alloys with high mechanical response are crucial for their practical usage. Yet, unifying the ZTE behavior and mechanical response in one material is a grand obstacle, especially in multicomponent ZTE alloys. Herein, we report a near isotropic zero thermal expansion ($\alpha_l = 1.10 \times 10^{-6}\,K^{-1}$, 260–310 K) in the natural heterogeneous $LaFe_{54}Co_{3.5}Si_{3.35}$ alloy, which exhibits a super-high toughness of $277.8 \pm 14.7\,J\,cm^{-3}$. Chemical partition, in the dual-phase structure, assumes the role of not only modulating thermal expansion through magnetic interaction but also enhancing mechanical properties via interface bonding. The comprehensive analysis reveals that the hierarchically synergistic enhancement among lattice, phase interface, and heterogeneous structure is significant for strong toughness. Our findings pave the way to tailor thermal expansion and obtain prominent mechanical properties in multicomponent alloys, which is essential to ultra-stable functional materials.

Zero thermal expansion (ZTE) in alloys, is generally manipulated by rigorous spin-lattice-orbital coupling[1–4], emerging in a variety of intermetallic compounds[5–9], etc. Such compounds tend to be brittle because of their multiple covalent or ionic bonds and the lack of independent slip systems[10–12]. As application-oriented, ZTE alloys must exhibit not only a low coefficient of thermal expansion ($\alpha_l \le 2.0 \times 10^{-6}\,K^{-1}$) but also good toughness to withstand external mechanical loads. A typical example is the ultra-stable components in gravitational wave detection (temperature windows ranging from 292.5–293.5 K)[13,14]. One effective approach to enhance mechanical performance is utilizing heterogeneous grain geometries with soft/hard-crossed structures[15–18], such as $(Mn, Fe)_2(P, Si)/Cu$[19], $La(Fe, Si)_{13}H_x/In$[20]. However, in chemically complex alloys (the case for most ZTE alloys), introducing a soft phase is not easy without compromising the thermal expansion performance of the hard matrix, like $Hf_{0.83}Ta_{0.13}Fe_{2+x}$[21], $La(Fe, Si)_{13}/\alpha$-Fe[22]. This is due to

unstoppable interface reactions that can quench the ZTE performance[21,23]. Particularly, to chemically complex alloys, element distribution and heterogeneous structure are difficult to control but vital to attaining isotropic ZTE and prominent mechanical responses.

$NaZn_{13}$-type $La(Fe, Si, Co)_{13}$ compounds, as typically multicomponent negative thermal expansion (NTE) alloys, are promising candidates for their cubic symmetry and colossal negative thermal expansion (NTE), such as $LaFe_{10.5}Co_{1.0}Si_{1.5}$ ($\alpha_l = -26.1 \times 10^{-6}\,K^{-1}$, 240–350 K)[6,24]. But they are extremely brittle and nearly devoid of ductility. To simultaneously meet both functional and mechanical goals, we aim to architect a natural heterogeneous structure of $La(Fe, Co, Si)_{13}/\alpha$-(Fe, Co, Si) (labeled as L/α phases). Hence, how to (i) maintain the giant NTE behavior of the L phase while introducing a ductile second α phase[22,25,26]; (ii) achieve precise regulation of the dual-phase content for zero thermal expansion (ZTE) performance[11,27] and (iii) assemble a well-constructed microstructure and interfacial

[1]Beijing Advanced Innovation Center for Materials Genome Engineering, and Institute of Solid State Chemistry, University of Science and Technology Beijing, Beijing 100083, China. [2]Institution of Physics, Chinese Academic of Science, No.8, 3rd South Street, Zhongguancun Haidian District, Beijing 100190, P. R. China. [3]Neutron Scattering Division, Oak Ridge National Laboratory, Oak Ridge, TN, USA. [4]X-Ray Science Division, Argonne National Laboratory, Argonne, IL 60439, USA. [5]Center of Electron Microscopy and State Key Laboratory of Silicon Materials, Department of Materials Science and Engineering, Zhejiang University, Hangzhou, China. [6]Guangxi Key Laboratory of Electrochemical and Magnetochemical Functional Materials, College of Chemistry and Bioengineering, Guilin University of Technology, Guilin 541004, P. R. China. ✉e-mail: xing@ustb.edu.cn

bonding to enhance mechanical behavior becomes the main factors[28,29]. These considerations were implemented as compositional design tactics.

Here, we demonstrate how constructing optimally composed heterostructures via chemical partitioning strategies. First, we observe that, for each element, there exists a nearly stable chemical partition between L and the second α phase, which is called the chemical partition coefficient here. Then, we establish chemical partition coefficients for silicon, iron, and cobalt in the L and α phases ($Si_L/Si_\alpha = 3.68$, $Fe_L/Fe_\alpha = 0.84$, $Co_L/Co_\alpha = 1.26$, the ratio of atomic percent in L and α phases) at an evaluated temperature (Supplementary Figs. 1, 2 and Methods). On this basis, heterogeneous alloys (L/α) are designed both facilely and precisely. Near room temperature, the $LaFe_{54}Co_{3.5}Si_{3.35}$ alloy exhibits an

isotropic ZTE behavior with super-high toughness ($277.8 \pm 14.7$ J·cm$^{-3}$). The appropriate crystal symmetry, chemical composition, and microstructure give rise to multi-property alloys.

## Results and discussion

### Thermal expansion performances and phase structure

Due to the chemical complexity in La-Fe-Co-Si alloys, systematic experiments have been conducted to build the chemical partition for regulating element distribution, which is highly correlated with magnetic ordering and interface microstructure (details in Methods and Supplementary Figs. 1, 2). The series $LaFe_{0.939x}Co_{0.061x}Si_{0.0583x}$ ($x = 37.5, 47.5, 57.5$, and $67.5$, labeled as S-1, S-2, S-3, and S-4) alloys were precisely designed and synthesized. Figure 1a shows the continuously

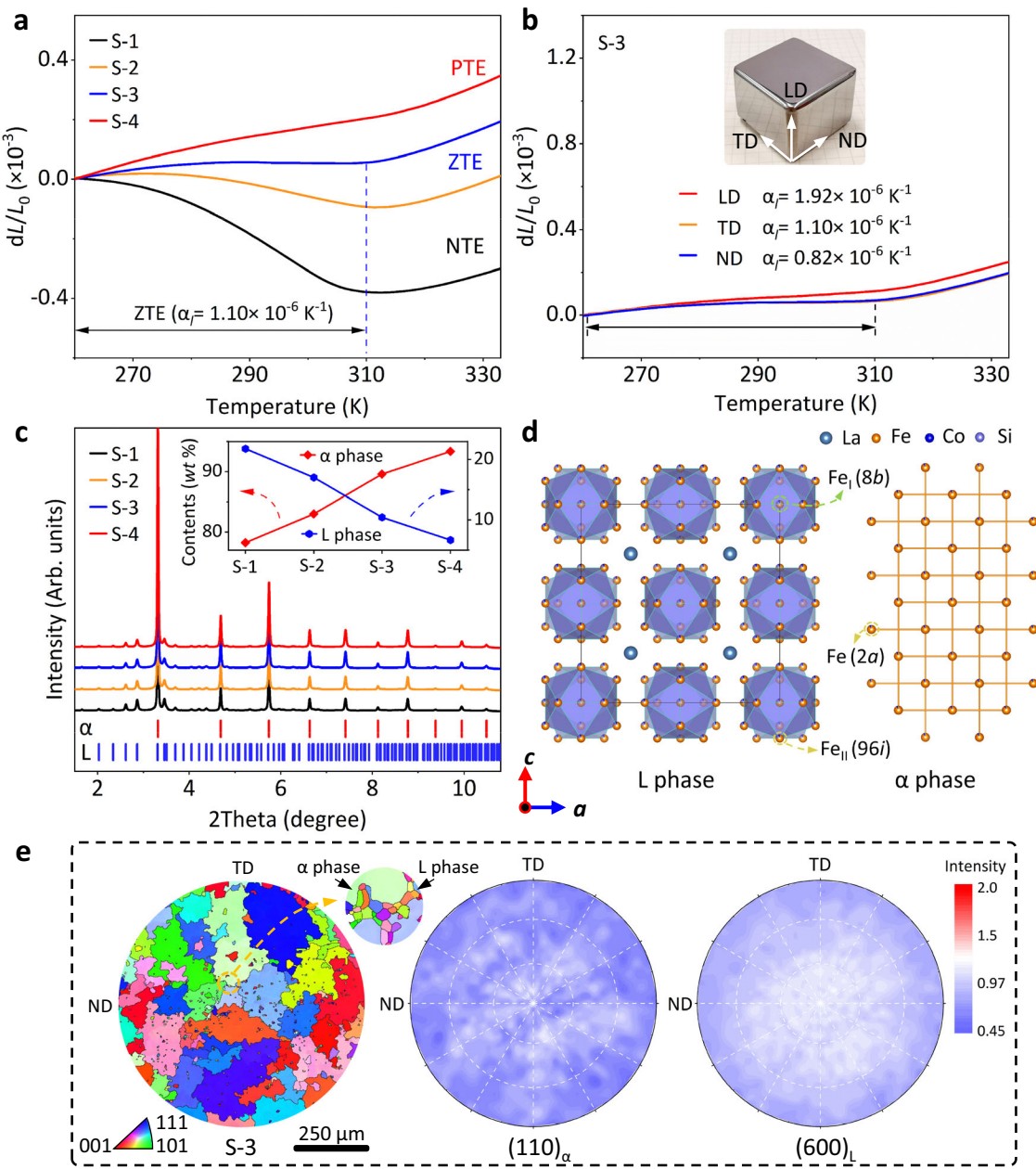

**Fig. 1 | The thermal expansion phase structure of the dual-phase alloys. a** The dilatometer thermal expansion behavior of $LaFe_{0.939x}Co_{0.061x}Si_{0.0583x}$ ($x = 37.5$, 47.5, 57.5, and 67.5, labeled as S−1, S-2, S-3, and S-4, respectively). **b** The dilatometer thermal expansion performance of S-3 (ZTE composition) alloy along loading direction (LD), transverse direction (TD), and normal direction (ND), respectively. **c** The synchrotron X-ray diffraction profiles of the as-cast samples (S−1, S-2, S-3, and

S-4), and the Bragg positions of the L and α phase are marked by blue and red vertical bars separately. Inserted the phase content (*wt* %) of L and α phases was determined by SXRD. **d** Crystal structures of L and α phase. **e** The EBSD inverse pole figure (IPFZ) and the neutron pole figure of the typical (*hkl*) reflections of the S-3 alloy, respectively.

tuned thermal expansion from negative thermal expansion (NTE) of S-1 ($\alpha_l$ = -8.23 × 10$^{-6}$ K$^{-1}$, 260–310 K) to positive thermal expansion (PTE) of S-4 ($\alpha_l$ = 4.12 × 10$^{-6}$ K$^{-1}$, 260–310 K). Impressively, a desirable ZTE was achieved in S-3 alloy that covers room temperature ($\alpha_l$ = 1.10 × 10$^{-6}$ K$^{-1}$, 260–310 K). More importantly, S-3 displays nearly isotropic ZTE behavior (Fig. 1b and Supplementary Fig. 3) and stable thermal shock resistance (Supplementary Fig. 4). Synchrotron X-ray diffraction (SXRD) reveals their dual-phase nature (Fig. 1c). The alloys comprise a body-centered-cubic (BCC, *Im-3m*) α-iron phase (Fe-Co-Si, labeled as α phase) with PTE and a face-centered-cubic (FCC, *Fm-3c*) phase (La(Fe, Co, Si)$_{13}$, labeled as L phase) with NTE (Supplementary Fig. 5). And the content of L phase form S-1 to S-4 was quantified to be 21.7 ± 0.2%, 17.0 ± 0.1%, 10.4 ± 0.1%, and 6.6 ± 0.2%, respectively (Inserted in Fig. 1c). From a crystallographic viewpoint, Fe and Co atoms would reside at three distinct crystallographic sites: Fe$_I$ (8b), Fe$_{II}$ (96i), and Fe (2a). Si atoms share the Fe$_{II}$ (96i) and Fe (2a) with Fe and Co atoms[30,31]. This occupation type results in deteriorating thermal expansion upon introduction of the α phase (Fig. 1d). However, based on chemical partitioning, both the chemical composition and phase structure of dual-phase alloys were regulated accurately. Besides, the bulk S-3 alloy exhibits an isotropic crystallographic orientation, which is further confirmed by the bulk neutron texture measurements of typical (110)$_\alpha$ and (600)$_L$ reflections (Fig. 1e and Supplementary Figs. 6, 7). The random crystallographic orientations ensure the near isotropy of its dilatometer thermal expansion.

## Microstructure and phase interface structure

The microstructure and phase interface of the composite are critical to the thermal shock resistance and mechanical response, as poor interfacial bonding can lead to fatigue failure[28,32,33]. Figure 2a, b clearly show

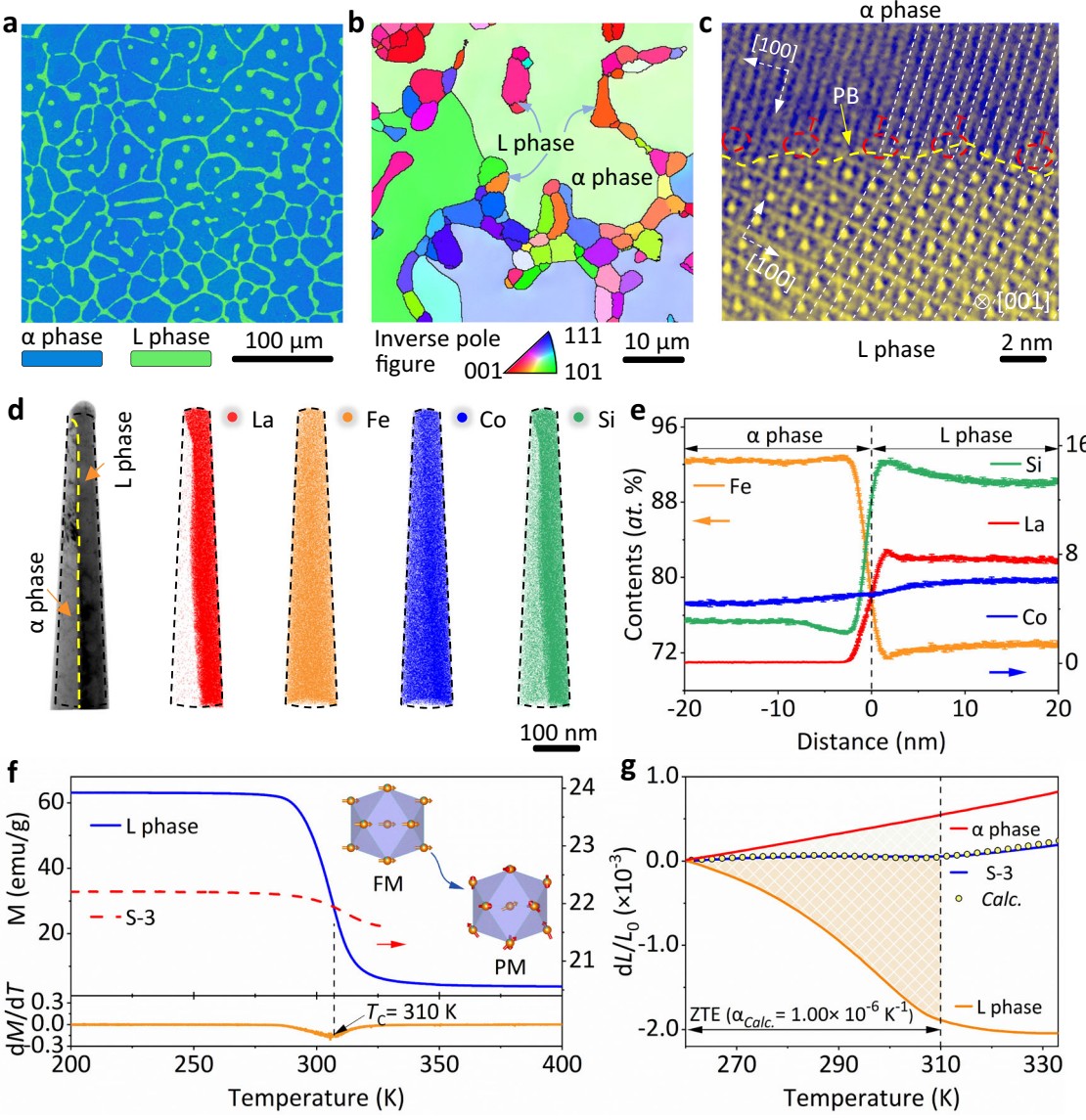

**Fig. 2 | The microstructure of the dual-phase alloys. a** Electro-probe micro-analyzer (EPMA) image of the S-3 alloy, L phase (green bar), and α phase (blue bar). **b** Electron back-scattered diffraction (EBSD) inverse pole map of the S-3 at high scales. **c** High-angle annular dark-field scanning TEM (HAADF-STEM) image of the corresponding phase interface along the [001]$_L$ zone axis. **d** Atom maps reconstructed using 3D-APT marked in nanoprobe with phase interface, α phase (left), and L phase (right). **e** One-dimensional concentration profile of the elemental distributions. The error bars represent the standard deviation. **f** Field-cooling (FC) magnetization (M) at a magnetic field of L phase (LaFe$_{10.30}$Co$_{0.83}$Si$_{1.87}$, 1000 Oe) and S-3 (500 Oe), respectively. Insert the magnetic structure model from ferromagnetic order (FM) to paramagnetic transition (PM). The Curie temperature ($T_C$) is determined by the d$M$/d$T$ curve. **g** The dilatometer thermal expansion of S-3, LaFe$_{10.30}$Co$_{0.83}$Si$_{1.87}$ (L phase), Fe$_{92.42}$Co$_{4.34}$Si$_{3.07}$ (α phase) alloys. The calculated S-3 ZTE (empty circle point line) is derived from L-phase and α phase thermal expansions.

the natural dual-phase heterostructure with the L phase (green bar, ~10 μm) precipitating at grain boundaries and inter-grains of the α phase (blue bar, ~200 μm). The elements uniformly distribute in the specific phases (Supplementary Fig. 8). Besides, a semi-coherent phase interface of the dual-phase alloy is observed (Fig. 2c and Supplementary Fig. 9), which may be attributed to the low structural mismatch between the two phases (Δ = 0.30 ± 0.01%). The edge dislocations are regularly arranged along $[100]_\alpha$ to relieve lattice stress from lattice misfit. Such a stable chemical interface bonding can modify phase boundaries[16,34], and transfer both thermal and lattice stress, resulting in stable thermal shock resistance.

Three-dimensional atom probe tomography (3D APT) has been employed to analyze the elemental distribution of the phase interface in S-3 alloy. The results demonstrated a clear chemical partition of elements, as evidenced by the varying enriched distributions of La, Fe, Si, and Co in Fig. 2d. The one-dimensional element distribution quantitatively describes the partition coefficient of elements in the dual phases (Fig. 2e): $Si_L/Si_\alpha = 4.34 ± 0.06$, $Fe_L/Fe_\alpha = 0.79 ± 0.06$ and $Co_L/Co_\alpha = 1.36 ± 0.07$, consistent with the results from electro-probe micro-analyzer (Supplementary Table 1 and Supplementary Table 2). As aforementioned, the exact chemical formula for the dual phases was determined by 3D-APT, $LaFe_{10.30}Co_{0.83}Si_{1.87}$ with NTE (L phase, $\alpha_l = -37.31 × 10^{-6} K^{-1}$, 260–310 K) and $Fe_{92.42}Co_{4.34}Si_{3.07}$ with PTE (α phase, $\alpha_l = 10.76 × 10^{-6} K^{-1}$, 260–310 K). Magnetic curves revealed that the NTE of the L phase ($LaFe_{10.30}Co_{0.83}Si_{1.87}$) was underpinned by ferromagnetic order to paramagnetic transition (Fig. 2f) and the ZTE performance in S-3 derives from L/α dual-phase composite (Fig. 2g, $\alpha_{Calc.} = 0.91 × 10^{-6} K^{-1}$, 260 – 310 K). These results show the strategy of the chemical partition can precisely construct the heterostructure with optimal chemical components. In addition, there was no segregation of deleterious elements (e.g., oxygen, nitrogen) at the interface, which validates the stability of the interface and highlights the superiority of natural heterostructures over conventional composites[35,36].

## Mechanical properties and strengthening mechanism

By architecting a natural heterostructure, we have effectively enhanced the mechanical strength and ductility of the dual-phase alloys, as demonstrated by the compressive stress-strain curves (Fig. 3a). Notably, the ZTE composition (S-3) exhibits a desirable ultimate strength ($\delta_{US}$) of 1.11 ± 0.03 GPa, and compressibility before fracture up to 30.9 ± 0.8 %. This significant improvement in toughness (277.8 ± 14.7 J·cm$^{-3}$, Supplementary Fig. 10a) enables the material to be easily fabricated into ultra-stable components for utilization in diverse fields (Supplementary Fig. 10b).

In situ neutron diffraction experiments elucidate the strengthening mechanisms of the dual-phase alloy (Fig. 3b and Supplementary Fig. 11)[11,33]. Based on the evolution of the lattice strain in typical {$hkl$} reflections (Fig. 3c), we identified three distinct stages: In stage I ($\delta < 0.2$ GPa), both the α phase and L phase underwent elastic deformation, during which they uniformly withstood the load. The yield strength ($\delta_{0.2}$) of the α phase is consistent with the single α-$Fe_{92.42}Co_{4.34}Si_{3.07}$ phase (Supplementary Fig. 12). In stage II ($0.2 < \delta < 0.6$ GPa), the α phase progressively yielded, while the L phase remained elastic. Meanwhile, the lattice strain of the L phase tuned upwards indicating stress transfer from the α phase to the L phase. The L phase supported most of the external loads in this stage. In stage III ($0.6$ GPa $< \delta$), the L phase began to yield. Interestingly, we observed a slowly increased lattice strain ($\varepsilon_{hkl}$) along the loading direction of the L

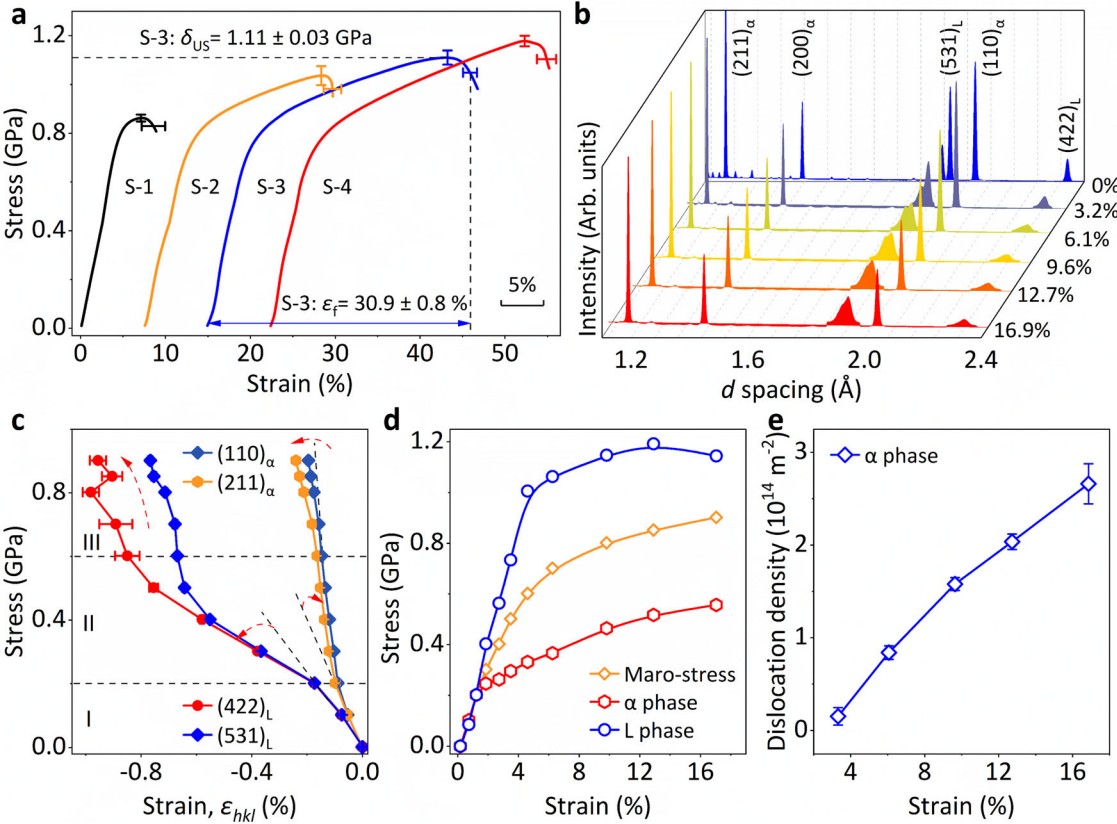

**Fig. 3 | Mechanical behaviors of the series dual-phase alloy. a** Engineering compressive stress-strain of the series dual-phase alloys. The error bars represent the standard deviation. **b** Neutron diffraction profiles at different strains ($\varepsilon$) along the loading direction (LD). The intensities of (531)$_L$ and (422)$_L$ peaks are magnified five times for a clear view. **c** lattice strain of the S-3 alloy along loading direction (LD). The error bars represent the standard deviation. **d** Macroscopic stress-strain compared with specific stress partitioning in α phase and L phase. Note that the strain is determined by in-situ neutron diffraction without an extensometer. **e** Dynamic dislocation densities ($\rho_\alpha$) of the α phase determined by modified Williamson-Hall (MWH) method. The error bars represent the standard deviation.

phase, demonstrating the L phase has a certain work-hardening behavior in the initial stage of plastic deformation. The deviated linear lattice strain ($\varepsilon_{hkl}$) response of the α phase reveals that the load is partly transferring from the L phase to the α phase in this stage (Fig. 3d). Further microstructural evolution analysis reveals a continuous initiation and accumulation of microcracks in the L phase, which is the predominant mode in this stage (Supplementary Fig. 13). Different from the movement and production of dislocations in the α phase (Fig. 3e), the generation of cracks in the L phase originates from the internal shearing of individual grains (Supplementary Fig. 13). The heterogeneous structure can restrain the crack propagation and make it have large uniform plastic deformation[18,37].

We further characterized the microstructure evolution of the dual-phase alloy by adopting TEM analysis[38-40]. It is noteworthy that during the initial stage of plastic deformation ($\varepsilon = 5\%$), the creation of defect structures in the L phase was observed (Supplementary Fig. 14). Given the forming mechanism for the defect structure, we selected two distinct regions to study their local structure at the atomic level (Fig. 4a). The defect structure is verified as a stacking fault that is formed by {111} <110> slip $\frac{\sqrt{2}}{2}a$ mode under the applied stress (Fig. 4b and

Supplementary Fig. 15). The slip mode was further corroborated by the observation of another region along the [110] crystal zone axis (Fig. 4c). The primary cause for the initial deformation is the stacking fault at the nanoscale. The formation of stacking faults may have contributed to its work-hardening behavior at the early stage of strain.

In contrast to the L phase, the dislocation movement and production of the α phase is the main deformation behavior, which benefits large plastic deformation (Fig. 4d–f). In turn, dislocations are gradually hindered by the L phase at the interface and packed to improve the strength. As the deformation continued, stacking faults in the L phase turned into shear cracks, which then evolved into microcracks (Fig. 4g–i). However, it is always confined in the L-phase grain and is obstructed when the crack propagates to the interface (Supplementary Fig. 16). The associated fracture morphology (Supplementary Fig. 17) provides evidence for the phase interfacial strengthening, displaying cross cracks and numerous dimples staggered unevenly. More specifically, a typical phase interface after large strain ($\varepsilon = 15\%$) was investigated (Supplementary Fig. 18). The interface remained well structural integrity, again demonstrating that interfacial

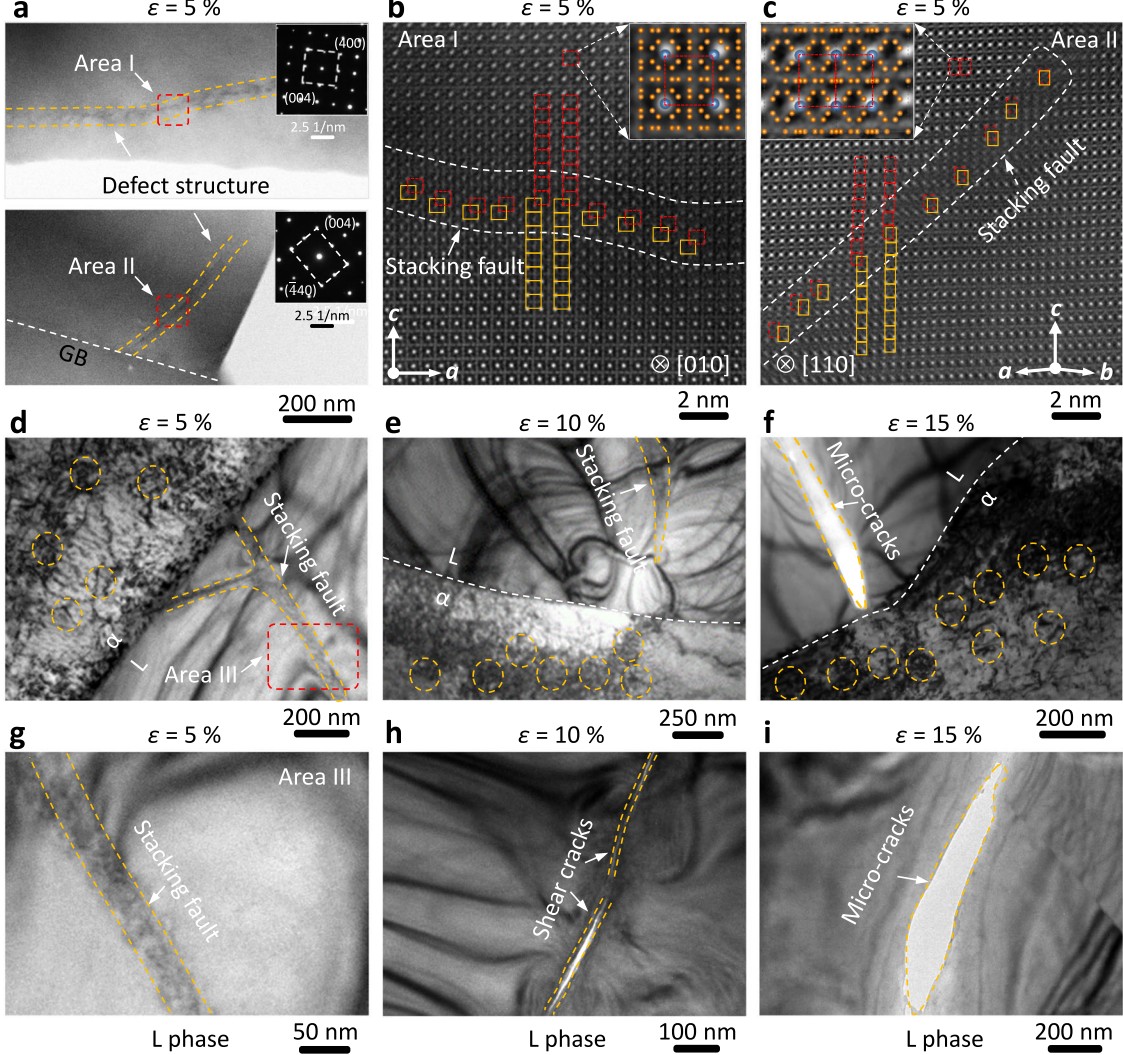

**Fig. 4 | Microstructural evolutions of the S-3 by TEM investigation. a** Defect structures in Area I and II of the L phase at $\varepsilon = 5\%$. **b, c** HAADF-STEM images of two typical stacking faults in areas I and II. Inserted the structure model of the L phase along different [010] and [110] orientations. **d–f** Microstructure evolutions at phase interfaces under different strains ($\varepsilon = 5\%$, 10%, and 15%). **g–i** Stacking faults evolutions in the L phase. Area III is marked by the red dashed box in (**d**). Noted: The structure of the cracks is affected to varying degrees by the thinning process.

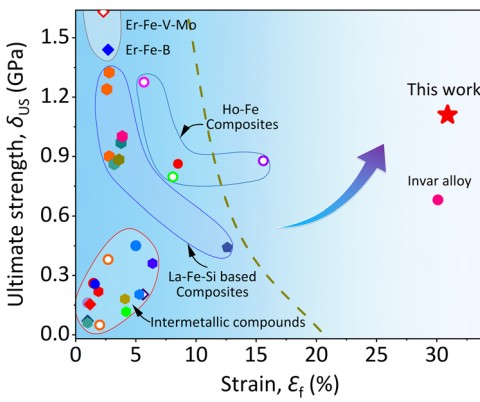

**Fig. 5 | Mechanical properties of ZTE alloys.** The comparison of the compressibility at fracture ($\varepsilon_f$) and ultimate strength ($\delta_{US}$) of various ZTE alloys[5,6,8,11,12,22,27,43–52]. The hollow and solid dots refer to anisotropic and isotropic thermal expansion, respectively. It is noted that Invar is a completely plastic material, for comparison, we used the compressive strength at $\varepsilon = 30\%$ strains here.

strengthening prevents intergranular fracture. Consequently, this hierarchically synergistic enhancement among lattice, phase interface, and heterogeneous structure gives rise to high toughness[37,41,42].

Figure 5 presents a comparison of the compressibility at fracture ($\varepsilon_f$) and ultimate strength ($\delta_{US}$) of various ZTE alloys (Supplementary Table 3)[5,6,8,11,22,27,43–52]. These intermetallic compounds have a natural tendency to be brittle. In our earlier work[11], we achieved an axial ZTE alloy in $Ho_{0.04}Fe_{0.96}$ with appropriate strength-plasticity ($\delta_{US} = 0.88$ GPa and $\varepsilon = 15.6\%$), but the anisotropic thermal expansion limited its applicability. Even in previously reported La(Fe, Si)$_{13}$-based composites[22,27,53], achieving excellent both strength and plasticity is scarce. This is due to the inadequate insight into chemical partition in chemically complex systems based on the chemical partition. In contrast, our present natural heterogeneous alloys are prominently distinguished among intermetallic ZTE alloys.

In summary, we developed the strategy of chemical partition to precisely synthesize natural heterogeneous alloys in the La-Fe-Co-Si quaternary system. An isotropic ZTE alloy ($LaFe_{54}Co_{3.5}Si_{3.35}$, $\alpha_l = 1.10 \times 10^{-6} K^{-1}$, 260–310 K) with an excellent toughness (277.8 ± 14.7 J·cm$^{-3}$) was designed and fabricated. Precise chemical regulation allowed for the harmonization of phase structure (L/$\alpha$ phase), chemical composition, and heterogeneous microstructure, thereby enabling controllable thermal expansion and mechanical responses. The hierarchically synergistic mechanism involving lattice symmetry, stable phase interface, and heterogeneous structure of dual phases played a crucial role in both the orderly transmission of stress and coordinated deformation of substructures, which are the origin of such an obvious mechanical response. This work provides a design route for reconciling the ZTE and mechanical properties in the chemically complex system, which is advantageous for the use of ZTE alloys.

## Methods
### Materials and preparations
The series of $LaFe_{0.939x}Co_{0.061x}Si_{0.0583x}$ ($x = 37.5, 47.5, 57.5,$ and $67.5$, labeled as S-1, S-2, S-3, and S-4, respectively) were prepared by arc melting of elements with purity better than 99.9% under a high-purity argon atmosphere. The samples were turned over and melted four times to ensure homogeneity. Then, the sample was followed by annealing at 1373 K in an argon atmosphere for about 24 h and quenched in liquid nitrogen (LN$_2$). The rods/tubes are melted under induction melting conditions and cast into Φ 25 mm diameter rods, annealed under vacuum (1373 K + 24 h), then machined to a suitable size and surface polished. Besides, to obtain the individual

dilatation curves for the $\alpha$ and L phases, we first determined the compositions of two phases ($Fe_{92.42}Co_{4.34}Si_{3.07}$ for the $\alpha$ phase and $LaFe_{10.30}Co_{0.83}Si_{1.87}$ for the L phase) based on the results of the 3D-APT. Second, we synthesized the single-phase $Fe_{92.42}Co_{4.34}Si_{3.07}$ ($\alpha$ phase) and $LaFe_{10.30}Co_{0.83}Si_{1.87}$ (L phase) in a targeted manner, respectively.

### Chemical partition coefficient determinations
The partition coefficients were calculated by the formula (1):

$$\text{Partition coefficient} = \frac{A_L, at.\%}{A_\alpha, at.\%} \tag{1}$$

Here, $A_L$, at.% and $A_\alpha$, at.% is the atomic percentage in the L phase and $\alpha$ phase, respectively. A represented Fe, Co, and Si elements.

First, we designed a series composition of $LaFe_{43.5}Si_x$ to endure the appropriate content of Si and Fe. The dilatometer thermal expansion reveals that $x = 2.6$ is suitable (Supplementary Fig. 1a). Thence, we tried to verify that the partition coefficient is a constant by the composition of $LaFe_{43.5+x}Si_{2.6+0.06x}$ ($x = 0, 4, 8, 12, 16$). We carefully selected three separate regions for the $\alpha$ and L phases of each component to measure their elemental compositions by the EPMA, as shown in Supplementary Fig. 1b and Supplementary Table 1. The ratio of $Si_L/Si_\alpha$ and $Fe_L/Fe_\alpha$ are sable at about 3.34 and 0.86 with negligible fluctuation. The dilatometer thermal expansion and magnetic measurement further confirmed the stability (Supplementary Fig. 1c, d). Second, we choose to partly replace Fe atoms with Co atoms in the component $LaFe_{57.5-x}Co_xSi_{3.35}$ ($x = 0, 1.5, 2.5, 3.5, 4.5$), the NTE behavior of the matrix can be successfully controlled, and ZTE is realized in $LaFe_{54}Co_{3.5}Si_{3.35}$. Considering that the Co element may also have a similar partition behavior, we also designed the $LaFe_{0.939x}Co_{0.061x}Si_{0.0583x}$ ($x = 37.5, 47.5, 57.5$ and $67.5$, labeled as S-1, S-2, S-3, and S-4, respectively) and test its compositional changes in dual phases (Supplementary Fig. 2 and Supplementary Table 2). A stable ratio of $Co_L/Co_\alpha$ (-1.26) was also demonstrated. Besides, the ratio of $Si_L/Si_\alpha$ and $Fe_L/Fe_\alpha$ in the La-Fe-Co-Si quaternary system is consistent with ternary La-Fe-Si system ($Si_L/Si_\alpha$ and $Fe_L/Fe_\alpha$ are 3.68, 0.84, respectively). The results further clarified the value of the partition coefficient is reasonable. The dilatometer thermal expansion and magnetic confirmed it (Supplementary Fig. 2c, d). As a result, we have achieved precise control by establishing partition coefficients between dual phases.

### Crystal and Microstructure characterization
The synchrotron X-ray diffraction (SXRD) of the samples at room temperature was collected in Advanced Photon Source (APS, $\lambda = 0.1173$ Å), USA. The elements mapping and the microstructure orientation of the samples were measured by using a scanning electron microscope (SEM, Zeiss Geminisem 500), and an electron backscattering diffraction device (EBSD, TESCAN MIRA 3 LMH SEM, and Symmetry EBSD). The brightfield images, SAED, and high-resolution transmission electron microscopy (HRTEM) were conducted at FEI Tecnai F30 transmission electron microscopy (TEM). The HAADF-STEM and ABF-STEM images were obtained on ARM 200CF (JEOL, Tokyo, Japan) transmission electron microscope operated at 200 kV and equipped with double spherical aberration (Cs) correctors. CAMECA Instruments LEAP 5000XR was used for the APT characterizations. The data were collected in voltage mode at a specimen temperature of 50 K, a pulse repetition rate of 200 kHz, a pulse fraction of 15%, and an ion collection rate of 0.5% per-field evaporation pulse. The APT data was reconstructed using Cameca IVAS 3.8.4, and the reconstruction was calibrated using crystallographic elements retained in the data as represented by spatial distribution

maps. The lattice mismatch of the two phases is based on the following formula (2):

$$\Delta = \frac{a_L - 4a_\alpha}{a_L} \times 100\%$$ (2)

## Magnetization measurements and dilatometer thermal expansion

The magnetization measurements were taken with a Quantum Design physical property measurement system (PPMS) equipped with a liquid helium-cooled vibrating sample magnetometer (VSM). The dilatometer thermal expansion was conducted by an advanced thermo-dilatometer (NETZSCH DIL402). The calculated dilatometer thermal expansion of the dual-phase alloy was calculated according to the formula (3) and (4):

$$\frac{dL}{L_0}(calc.) = \frac{dL}{L_0}(\alpha) \times Vol._\alpha\% + \frac{dL}{L_0}(L) \times Vol._L\%$$ (3)

$$\alpha l = \frac{dL}{L_0}/\Delta T$$ (4)

Here, $\frac{dL}{L_0}$ is the dimensional change across the temperature interval $\Delta T$. $Vol.$ is the volume percentage. $\alpha_l$ represents the coefficient of linear expansion.

## Mechanical properties

The compressive stress-strain curves were measured using a CMT4105 universal electronic machine at room temperature, with a $\Phi$ 5 × 10 mm cylinder and an initial strain rate of $1.0 \times 10^{-3} s^{-1}$. Each sample was tested a total of four times.

## In situ neutron diffraction measurements

The three-dimensional crystallographic texture and in-situ loading study by neutron diffraction were performed at the VULCAN beamline (BL-7) in Oak Ridge National Laboratory (ORNL), USA. The analysis of neutron data was based on VDRIVE software and GASA software.

## Lattice strain and phase-specific stress calculations

The single peak fitting method was used to determine the lattice strain of the specific ($h$ $k$ $l$) reflections during loading. The lattice strain was calculated using the formula (5)

$$Strain = \frac{(d_1 - d_0)}{d_0} \times 100\%$$ (5)

Here, $d_1$ and $d_O$ represent the interplanar crystal spacing of the ($h$ $k$ $l$) crystal plane after and before loading, respectively. For average lattice strain ($\varepsilon_i$), the $d_1$ and $d_O$ are replaced by the unit cell parameters ($a_1$ and $a_O$). The phase-specific stress was calculated by following formula (6):

$$\sigma_i = \frac{E_i}{(1+v_i)(1-2v_i)} \times \{(1-v_i) \times \varepsilon_{i,11} + v_i \times (\varepsilon_{i,22} + \varepsilon_{i,33})\} + \sigma_r$$ (6)

where $i$ stands for $\alpha$ and L, $\sigma_i$ is the stress in the loading direction, $E_i$ is the diffraction elastic modulus, $v_i$ is the Poisson's ratio, $\sigma_r$ is the thermal residual stress, $\varepsilon_{i,11}$ is the lattice strain in LD, $\varepsilon_{i,22}$ and $\varepsilon_{i,33}$ are the lattice strains in TD and ND, respectively. The $\varepsilon_{i,22} = \varepsilon_{i,33}$ and can be measured by TD.

## Dynamic dislocation density calculation

The modified *Williamson-Hall* (MWH) method was used to calculate dislocation density, which quantifies the effects of average sub-grain (crystallite) size $D$ and micro-strain fields on line broadening as formula (7):

$$(\Delta K)^2 = (0.9/D)^2 + (\pi A^2 b^2/2\rho) \times (K^2\bar{C}) + O(K^2\bar{C})^2$$ (7)

Here, $K = 1/d$, $\Delta K = -K(\Delta d/d)$, $d$ is the interplanar spacing, $\Delta d$ is FWHM, $b$ is Burgers vector length of the dislocation, $\rho$ is dislocation density, $A$ is a constant that depends on the effective outer cutoff radius of dislocations, besides $O(K^2\bar{C})$ represents the non-interpreted high-order terms. $\bar{C}$ is the average dislocation contrast factor.

## Data availability

The data that support the findings of this study are available from the corresponding authors upon request.

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

## Acknowledgements

This research was supported by National Key R&D Program of China (2020YFA0406202) (X.R.X.), National Natural Science Foundation of China (22090042 (X.R.X.) and 21971009 (K.L.)), Guangxi BaGui Scholars Special Funding, China Postdoctoral Science Foundation (2023M740210) (C.Y.Y.), and the Fundamental Research Funds for the Central Universities, China (FRF-IDRY-GD21-03, GJRC2023003, and FRF-EYIT-23-03) (K.L.). The synchrotron radiation experiments were performed in Advanced Photon Source (APS, λ = 0.1173 Å), USA; Neutron diffraction work was carried out at the Spallation Neutron Source (SNS) (Proposal No. 2022A30005.1), which is the U.S. Department of Energy (DOE) user facility at the Oak Ridge National Laboratory, sponsored by the Scientific User Facilities Division, Office of Basic Energy Sciences.

## Author contributions

X.X. and C.Y. conceived the idea of the work and supervised the project. C.Y. synthesized the alloys. C.Y. carried out the main experiments. K.L. and T.L. analyzed the SXRD data. H.Z. conducted the 3D APT experiment. Y.Ca. helped with the measurements of magnetism. X.K., J.D. and Q.L. analyzed the thermal expansion results. Q.Z., L.Y., X. F. and Q.Y. conducted the TEM measurements. Y.Ch., K.A. and D.Y. analyzed the in-situ neutron diffraction results. All authors discussed the results and commented on the manuscript.

## Competing interests

The authors declare that they have no conflict interests.
