## [Peer Review File · Nature Communications]

An isotropic zero thermal expansion alloy with super-high toughnessREVIEWER COMMENTS

Reviewer #1 (Remarks to the Author):

The present manuscript deals with the zero thermal expansion and mechanical properties in La-Fe-Co-Si quaternary alloys. The method of designing the ZTE material with chemical partition is very interesting and impressive. The results are good, but there are still a few issues that should be addressed to improve the quality of manuscript:

(1) The author claims that "More importantly, S-3 displays nearly isotropic ZTE behavior and stable thermal shock resistance (Fig. 1b and Supplementary Fig. 3)" in lines 80-81. How to understand "stable thermal shock resistance"? I think this should include two parts: thermal expansion and mechanical properties. Supplementary mechanical property testing after thermal cycling is needed to prove the stability of the mechanical properties.

(2) In Fig. 1e, the grain size of the dual-phase alloy seems to be very different. The α phase (200 μm) and the L phase are only about 10 μm . why? In addition, since EBSD data can derive pole figures, why is it only necessary to measure the crystallographic texture with neutrons?

(3) The author claims "The microstructure and phase interface of the composite is critical to the thermal shock resistance and mechanical response, as poor interfacial bonding can lead to fatigue failure" in lines 106-107, Page 6. I am very curious whether the interface structure contributes to the thermal expansion properties of dual-phase alloys.

(4) In lines 118-121, Page 6, there seems to be a certain deviation in the distribution coefficient values measured using 3D APT and EPMA. For example, $\text{SiL}/\text{Si}\alpha = 4.34 \pm 0.06$ (APT) and 3.68 (EPMA). What is the reason for this error?

(5) The compressive stress-strain curves in Fig. 3d look too soft. Is this because there is no extensometer? This requires a reasonable explanation.

Reviewer #2 (Remarks to the Author):

I have examined the article "An isotropic zero thermal expansion alloy with super-high toughness " by Yu et al. The manuscript is well written through detailed representation, SXRD, TEM, APT, NPD, etc., and achieved an isotropic zero thermal expansion ($\alpha=1.10 \times 10^{-6}/\text{K}$, 260-310 K) in La-Fe-Co-Si alloy with super-high toughness of $277.8 \pm 14.7 \text{ J}\cdot\text{cm}^{-3}$ via the strategy of the chemical partition. Some points deserve further clarification and discussion:

1. I noticed that the author has reported a superior ZTE alloy in the Er-Fe-B system (Ref: 49 of the manuscript) by boron-migration mediated solid-state reaction. Why is the synthetic strategy of chemical partitioning proposed in this work? Are there any intrinsic connections and differences between the two?

2. In Fig. 2c, there seems to be a certain degree of coherence between the two phases. Is this common? The two-phase interface relationship and the location of edge dislocation should be described in detail from the perspective of lattice parameters.

3. According to the results of in-situ NPD, α phase yields at $\sim 200 \text{ MPa}$. Has the stress-strain curve of the pure α phase been measured? In other words, has the α phase been strengthened by the L phase?

4. The formula (4) looks like there is one less equal sign (=).

5. In Fig. 4a-c, the author claimed "The defect structure is verified as a stacking fault that is formed by $\{111\} \langle 110 \rangle$ slip $\sqrt{2}/2a$ mode under the applied stress (Fig. 4b)." This is difficult to understand, and a schematic diagram of the crystal structure needs to be drawn to support it.

6. It may be worth considering replacing Fig. 5b in the Supplementary Information to make the article more academic.

7. The Supplementary Information needs to be checked and modified, for example, "GASA" should be "GSAS".

In conclusion, I believe that the topic raised by the authors is relevant and of potential interest to a wide community of material researchers. The results are interesting, and they are presented clearly.

However, several issues should be answered before considering publication.

Reviewer #3 (Remarks to the Author):

This paper is certainly interesting and deserves dissemination. The following points should be addressed first:

I37: Authors state that "As application-oriented, ZTE alloys must exhibit not only an extremely low coefficient of thermal expansion ($\alpha \leq 2.0 \times 10^{-6} \text{ K}^{-1}$) but also excellent toughness to withstand external mechanical loads." Which external loads are authors referring to? Authors should mention specific applications that require very high toughness ZTE materials.

I82-I85 and Fig. 1b: Because of their symmetry, cubic phases have isotropic thermal expansion (independent of whether positive, zero, or negative). Hence, combining two cubic phases results in isotropic thermal expansion, which is also expected for the dual-phase microstructures studied here. However, their thermal expansion is somewhat anisotropic (Fig. 1b). Where do these variations between different sample directions in Fig. 1b come from?

I86: Authors write that "From a lattice viewpoint, there were three crystallographic sites (FeI (8b), FeII (96i), and Fe (2a)) arbitrarily occupied by Fe, Co, and Si atoms". How do authors reach this conclusion? Via measurements (which?), theoretical calculations (which?). This should be explained.

Fig. 1e and Fig. 2a: The microstructures in these figures look very different, why? The interdendritic L phase visible in Fig. 2a (mostly continuous network) seems to have a different morphology in Fig. 1a (individual islands).

Fig. 1b: How do LD, TD, ND relate to the ingot directions?

I130: Specify what is meant by "natural heterostructures" and "conventional composites". What are their characteristics?

Fig. 2g: How were the individual dilatation curves for alpha and L phases obtained? A dilatometer is mentioned in the caption, but how can authors extract expansion curves for individual phases from a dilatometer curve measured on a dual-phase polycrystal? Details must be provided otherwise the analysis remains unreasonable.

I163: "Further microstructural evolution analysis reveals a continuous initiation and accumulation of microcracks in the L phase, which is the predominant mode in this stage (Supplementary Fig. 10)." Check the reference to Supp. Fig. 10, likely Supp. Fig. 11 is meant.

Fig. 3d,e: Why are data only shown in strain range 0 to 16% even though the uniform elongation is almost 31% (Fig. 3a)?

I181: Explain what is meant by "deep mechanism".

I195: Authors claim that dislocations in alpha phase belong to slip system $\{110\} \langle 111 \rangle$, however do not show slip trace or Burgers vector analysis. Hence, the authors should revise their statement or provide a dislocation analysis.

I199: See comment about Supp. Fig. 12 d, f.

Fig. 5a: Classical Fe-Ni based Invar alloys should be included in the graph, since they are the industry standard and widely used.

Linked to that, authors should briefly discuss the magnetization of the present alloys in relation to Fe-Ni Invar alloys. Can the present alloys be used in magnetic fields?

Fig. 5b: Authors should explain the fabrication process of the rods/tubes shown.

Explain the "Counts" axis in Supp. Fig. 5. What does it represent?

I289: Authors write that dL/L_0 is the dimensional change per unit temperature. If so, then there would be no need to divide dL/L_0 by ΔT in (3). More likely, dL/L_0 is simply the relative length change across the temperature interval ΔT .

I291: How was strain measured?

I301: Eq. (4) misses an equal sign.

Supp. Fig. 12 a,b,c: Authors should provide clean and sharp images of the stacking faults typically showing fringes when viewed at an inclination angle (for instance Fig. 2 in <https://normandie-univ.hal.science/hal-02175446>).

Supp. Fig. 12 d, f: Microcracks in f seem to be holes originating from TEM lamella thinning process. Also, for two "microcracks" suggested by yellow dashes in d and f no cracks are seen in the images.

Language:

The sentence in I34 misses a verb.

I36: slip systems of independence ==> independent slip systems

I239: Authors write: "...arc melting with the element more than 99.9% under high purity argon..." Do they mean "...arc melting of elements with purity better than 99.9% under high purity argon..."?

I281: Delete the single "a" in thermal expansion a.

The sentence starting on I291 is incomplete.

Exploring zero thermal expansion (ZTE) alloys with exceptional mechanical response is crucial for their practical use. \ Zero thermal expansion (ZTE) alloys with exceptional mechanical response are crucial for practical applications.

Reviewer #1 (Remarks to the Author):

The present manuscript deals with the zero thermal expansion and mechanical properties in La-Fe-Co-Si quaternary alloys. The method of designing the ZTE material with chemical partition is very interesting and impressive. The results are good, but there are still a few issues that should be addressed to improve the quality of the manuscript:

Reply: We thank the reviewer for their careful reading of our manuscript and for acknowledging the interest and significance. We also appreciate your insightful comments and constructive suggestions. Following these comments/suggestions, we have (i) conducted the thermal expansion cycle stability and mechanical property stability to verify the “stable thermal shock resistance”; (ii) added the comparison of EBSD and neutron pole figure; (iii) provided an additional discussion of the thermal expansion behavior of interface structures and (v) revised some mistakes. *Based on these results, we have completely revised the manuscript. The detailed corrections are listed below.*

Comments 1: *The author claims that “More importantly, S-3 displays nearly isotropic ZTE behavior and stable thermal shock resistance (Fig. 1b and Supplementary Fig. 3)” in lines 80-81. How to understand “stable thermal shock resistance”? I think this should include two parts: thermal expansion and mechanical properties. Supplementary mechanical property testing after thermal cycling is needed to prove the stability of the mechanical properties.*

Reply: (i) For composite materials, especially ZTE materials that resist temperature fluctuations, performance stability during service is very critical. “Stable thermal shock resistance” should include ZTE and mechanical properties. (ii) Therefore, we supplemented the thermal expansion cycle stability and mechanical property stability tests in the revised manuscript (Fig. R1). The results show that the present ZTE alloy (S-3) keeps ZTE and the mechanical properties stable after experiencing 200 thermal cycle tests. We have added a detailed discussion in the revised Supplementary information.

Fig. R1 The cyclic thermal shock experiment undergoes a thermal shock from 77 K to 373 K. **a**, The dilatometer thermal expansions of S-3 alloy in the 1st, 100th, and 200th cycles. **b**, The compressive stress-strain curves of the S-3 alloy after the 1st, 100th, and 200th cycles.

Comments 2: *In Fig. 1e, the grain size of the dual-phase alloy seems to be very different. The α phase (200 μm) and the L phase are only about 10 μm . why? In addition, since EBSD data can derive pole figures, why is it only necessary to measure the crystallographic texture with neutrons?*

Reply: (i) This is due to the difference in the formation mechanisms of the two phases. The α phase is generated in large quantities during the liquid phase solidification process. However, the L phase requires long aging at a high temperature to fully precipitate. Therefore, the grain size of the L phase is smaller. (ii) Crystallographic texture has a strong influence on dilatometer thermal expansion properties. Although EBSD data can obtain pole figures, it counts a small number of grains and can only obtain surface information. In comparison, neutron diffraction has a strong penetrating ability and can obtain crystallographic information of the entire sample, and the results are more accurate. As shown in Figure R2, there seems to be a certain crystallographic texture of the L phase, which was tested by EBSD. This was caused by the limitations of the EBSD, so we conducted additional neutron diffraction tests. We have added a detailed discussion in the revised Supplementary information.

Fig. R2 a, b The EBSD pole figure (a) and the neutron pole figure (b) of the $(110)_\alpha$, $(600)_L$ reflections of the S-3 alloy, respectively.

Comment 3: *The author claims “The microstructure and phase interface of the composite is critical to the thermal shock resistance and mechanical response, as poor interfacial bonding can lead to fatigue failure” in lines 106-107, Page 6. I am very curious whether the interface structure contributes to the thermal expansion properties of dual-phase alloys.*

Reply: The interface structure should have an impact on thermal expansion. Because the essence of thermal expansion is that the non-harmonic vibration of phonons causes atoms to deviate from their equilibrium position. In our previous work, we reported the twin crystal-induced near-zero thermal expansion in SnO_2 nanowires ^[1], which can confirm the interface does have an impact on thermal expansion. However, in bulk dual-phase alloy, the interface is crucial to the stability of thermal expansion, but the impact on the magnitude of thermal expansion needs to be discussed.

Based on the results of the 3D-APT, we synthesized the single-phase $\text{Fe}_{92.42}\text{Co}_{4.34}\text{Si}_{3.07}$ (α phase) and $\text{LaFe}_{10.30}\text{Co}_{0.83}\text{Si}_{1.87}$ (L phase) in a targeted manner, respectively. And tested their dilatometer thermal expansion, as shown

in Fig. R3. The L phase shows negative thermal expansion ($\alpha_l = -37.31 \times 10^{-6} \text{ K}^{-1}$, 260 - 310 K), and the L phase shows positive thermal expansion ($\alpha_l = 10.76 \times 10^{-6} \text{ K}^{-1}$, 260 - 310 K). Based on this, we calculated the dilatometer thermal expansion of the S-3 as the following formulas (1) and (2):

$$\frac{dL}{L_0}(\text{calc.}) = \frac{dL}{L_0}(\alpha) \times \text{Vol.}_\alpha \% + \frac{dL}{L_0}(L) \times \text{Vol.}_L \% \quad (1)$$

$$\alpha_{\text{calc.}} = \frac{dL}{L_0}(\text{calc.})/\Delta T \quad (2)$$

As a result, the coefficient of calculated thermal expansion ($\alpha_{\text{calc.}}$) is $0.91 \times 10^{-6} \text{ K}^{-1}$ (260 - 310 K), which is almost consistent with the dilatometer thermal expansion of S-3 alloy ($\alpha_l = 1.00 \times 10^{-6} \text{ K}^{-1}$, 260 - 310 K). Therefore, we believe that the interface structure has little influence on the dilatometer thermal expansion of current dual-phase alloys, and the phase structure and phase content dominate it.

Fig. R3 The dilatometer thermal expansion of S-3, $\text{LaFe}_{10.30}\text{Co}_{0.83}\text{Si}_{1.87}$ (L phase), $\text{Fe}_{92.42}\text{Co}_{4.34}\text{Si}_{3.07}$ (α phase) alloys. The calculated S-3 ZTE (empty circle point line) is derived from L-phase and α phase thermal expansions.

[1] Zhu H., *et al.* Twin crystal induced near zero thermal expansion in SnO_2 nanowires. *J. Am. Chem. Soc.* 140, 7403-7406 (2018).

Comment 4: *In lines 118-121, Page 6, there seems to be a certain deviation in the distribution coefficient values measured using 3D APT and EPMA. For example, $\text{Si}_l/\text{Si}_\alpha = 4.34 \pm 0.06$ (APT) and 3.68 (EPMA). What is the reason for this error?*

Reply: This is due to the different accuracy of the two test instruments. The

elemental composition is obtained through point analysis of the electro-probe micro-analyzer (EPMA). However, the results of 3D-APT are reconstructed through mass spectrometry, and its elemental composition is calculated from the data of the entire probe. Although their absolute values differ, the trends are reasonable.

Comment 5: *The compressive stress-strain curves in Fig. 3d look too soft. Is this because there is no extensometer? This requires a reasonable explanation.*

Reply: The stress-strain curves in Fig. 3d are measured by in-situ loading neutron diffraction without an extensometer. This is because adding an extensometer will bring redundant diffraction signals and affect the collection of neutron data. We have added a detailed explanation in the revised manuscript.

Reviewer #2 (Remarks to the Author):

I have examined the article "An isotropic zero thermal expansion alloy with super-high toughness " by Yu et al. The manuscript is well written through detailed representation, SXRD, TEM, APT, NPD, etc., and achieved an isotropic zero thermal expansion ($\alpha=1.10\times 10^{-6}/K$, 260-310 K) in La-Fe-Co-Si alloy with super-high toughness of $277.8 \pm 14.7 \text{ J}\cdot\text{cm}^{-3}$ via the strategy of the chemical partition. Some points deserve further clarification and discussion

Reply: We appreciate the referee's positive feedback and constructive comments that we addressed. In response to these comments/suggestions, we have (i) conducted a detailed discussion between the current work and the Er-Fe-B alloy; (ii) selected two interfaces to conduct STEM measurement, which further verify the relationship between the two phases at the phase interfaces. (iii) supplemented the crystal model of the stacking faults and (iv) checked the manuscript carefully. *We have revised the entire manuscript. The corrections are enumerated in detail below.*

Comment 1: *I noticed that the author has reported a superior ZTE alloy in the Er-Fe-B system (Ref: 49 of the manuscript) by boron-migration mediated solid-state reaction. Why is the synthetic strategy of chemical partitioning proposed in this work? Are there any intrinsic connections and differences between the two?*

Reply: Our research (both in previous work in *Nat. Commun.* 14, 3135, 2023 and in this work) is application-oriented, aiming to strengthen and toughen zero thermal expansion intermetallic compounds so that they can be truly applied. But there are essential differences between the two works:

(i) **The key properties are different.** In our previous work, "Superior zero thermal expansion dual-phase alloy via boron-migration mediated solid-state reaction" the compressive strength is high but no plasticity (Fig. R4 a). However, excellent plasticity is important for the service stability and processability of the material. Therefore, we focus on solving it in the next work. As a systematic in-

depth study, we found through the strategy of chemical partition to precisely synthesize natural heterogeneous alloys in the La-Fe-Co-Si quaternary system. An isotropic ZTE alloy ($\text{LaFe}_{54}\text{Co}_{3.5}\text{Si}_{3.35}$, $\alpha_l = 1.10 \times 10^{-6} \text{ K}^{-1}$, 260 - 310 K) with an exceptional toughness ($\delta_{US} = 1.11 \pm 0.03 \text{ GPa}$, $\varepsilon_f = 30.9 \pm 0.8 \%$) was designed and fabricated (Fig. R4 b), which is more challenging and desired in practical usage.

Fig. R4 Engineering compressive stress-strain curves of the Er-Fe-B ZTE alloys (a) and La-Fe-Si dual-phase alloys (b), respectively.

(ii) **The design principles are different.** In our previous work (*Nat. Commun.* 14, 3135, 2023), we synthesized the dual-phase alloy by boron-migration mediated **solid-state reaction** in Er-Fe-B ternary system (Fig. R5 a). From a structural point of view, the three atomic radii ($r_{\text{Er}} = 1.75 \text{ \AA}$, $r_{\text{Fe}} = 1.27 \text{ \AA}$, $r_{\text{B}} = 0.85 \text{ \AA}$) are very different and occupy their specific sublattice (Fig. R5 b). In the process of introducing the second phase (α -Fe), there will be no diffusion of interface elements and deterioration of the thermal expansion performance of the matrix phase ($\text{Er}_2\text{Fe}_{14}\text{B}$). However, In the La-Fe-Si ternary alloy system, there were three crystallographic sites (Fe_I (8b), Fe_{II} (96i), and Fe (2a))

arbitrarily occupied by Fe, Co, and Si atoms, resulting in deteriorating thermal expansion upon introduction of the α phase (Fig. R5 b). To face the dilemma head-on, we proposed the strategy of chemical partition in a chemical complex system, which assumes the role of not only modulating thermal expansion through magnetic interaction but also enhancing mechanical properties via interface bonding. We believe the present results are novel and represent important progress in the society of ZTE functional materials and are highly instructive for follow-up work.

Fig. R5 a, The schematic diagram of boron-migration-mediated solid-state reaction. **b**, Crystal structures of $\text{Er}_2\text{Fe}_{14}\text{B}$, L phase, and α phase.

Comment 2: *In Fig. 2c, there seems to be a certain degree of coherence between the two phases. Is this common? The two-phase interface relationship and the location of edge dislocation should be described in detail from the perspective of lattice parameters.*

Reply: (i) The interface orientation relationship between the two phases is diverse, as shown in the results of EBSD IPFZ (Fig. R6). And the coherent interfaces are common here. To further verify it, we have selected two interfaces

to conduct STEM measurement (Fig. R7). It reveals that there is a phase relationship between the two phases at the phase interfaces. (ii) We have calculated the lattice mismatch of the two phases based on the following formula (1):

$$\Delta = \frac{a_L - 4a_\alpha}{a_L} \times 100 \% \quad (1)$$

Added the discussion of the two-phase interface relationship and the location of edge dislocation in the revised manuscript: “Besides, a semi-coherent phase interface of the dual-phase alloy is observed (Fig. 2c), which may be attributed to the low structural mismatch between two phases ($\Delta = 0.30 \pm 0.01 \%$). The edge dislocations are regularly arranged along $[100]_\alpha$ to relieve lattice stress from lattice misfit.”

Fig. R6 a-c, The EBSD band contrast figure at different magnifications. **d-f**, The EBSD inverse pole figure (IPFZ) at different magnifications.

Fig. R7 a-d, The phase interfaces of the alloy and corresponding selected area electron diffraction (SAED). **e-l**, The HAADF-STEM images at the phase interface.

Comment 3: According to the results of in-situ NPD, α phase yields at ~200 MPa. Has the stress-strain curve of the pure α phase been measured? In other words, has the α phase been strengthened by the L phase?

Reply: We have supplemented the compressive stress-strain curves of the pure L phase ($\text{LaFe}_{10.30}\text{Co}_{0.83}\text{Si}_{1.87}$) and α phase ($\text{Fe}_{92.42}\text{Co}_{4.34}\text{Si}_{3.07}$). The pure α phase is yielded at $\delta_{0.2} = 204$ MPa (Fig. R8) and there is no obvious enhancement by the L phase.

Fig. R8 The compressive stress-strain curves of the pure L phase ($\text{LaFe}_{10.30}\text{Co}_{0.83}\text{Si}_{1.87}$) and α phase ($\text{Fe}_{92.42}\text{Co}_{4.34}\text{Si}_{3.07}$).

Comment 4: *The formula (4) looks like there is one less equal sign (=).*

Reply: Sorry for the mistake, we have added it in the revised manuscript.

Comment 5: *In Fig. 4a-c, the author claimed “The defect structure is verified as a stacking fault that is formed by $\{111\} \langle 110 \rangle$ slip $\sqrt{2}/2a$ mode under the applied stress (Fig. 4b).” This is difficult to understand, and a schematic diagram of the crystal structure needs to be drawn to support it.*

Reply: Thanks for your constructive comments and suggestions. To better understand the evolution of stacking faults, I have added a structural model in the revised Supplementary information (Fig. R9).

Fig. R9 The evolution of stacking fault illustrated by the crystal structure. To

make the model clearer, we only keep La atoms here.

Comment 6: *It may be worth considering replacing Fig.5b in the Supplementary Information to make the article more academic.*

Reply: We have replaced the Fig.5b in the Supplementary Information.

Comment 7: *The Supplementary Information needs to be checked and modified, for example, "GASA" should be "GSAS".*

Reply: Sorry for the mistake, we have carefully checked the supplementary information in the revised manuscript.

Reviewer #3 (Remarks to the Author):

This paper is certainly interesting and deserves dissemination. The following points should be addressed first:

Reply: We are pleased to know that the referee found our work interesting and thank you for the constructive comments, which we addressed below. Especially we would like to thank the referee for the time dedicated to correct grammar and language. In response to these comments/suggestions, we have (i) conducted EMPA experiments to verify the slight difference in the thermal expansion originating from the microstructure along different directions; (ii) checked the occupation of Fe, Co, and Si atoms in FeI (8b), FeII (96i), and Fe (2a) sub-lattice; (iii) re-conducted the EBSD experiment of the S-3 alloy to confirm the microstructural consistency; (iv) supplemented the explanation of the individual dilatometer thermal expansion curves for the α and L phases were obtained based on the results of the 3D-APT; (v) added the Invar alloy in Fig. 5a and conducted the magnetic curve of Invar alloy; (vi) supplemented the explanation of the neutron texture profiles in the supplementary information; (vii) conducted TEM measurements to verify the microstructure of the stacking fault. (viii) conducted the SEM at 15% strain to determine whether the shear cracks were indeed observed in the L phase. (IX) revised the grammar and language carefully. *Based on the results of these experiments and supplementary, we revised the manuscript extensively. Below is a summary of the specific corrections.*

Comment 1: *I37: Authors state that "As application-oriented, ZTE alloys must exhibit not only an extremely low coefficient of thermal expansion ($\alpha_l \leq 2.0 \times 10^{-6} K^{-1}$) but also excellent toughness to withstand external mechanical loads." Which external loads are authors referring to? Authors should mention specific applications that require very high-toughness ZTE materials.*

Reply: This is because we noticed that ZTE intermetallic compounds, as a main

branch of ZTE materials, exhibit intrinsic brittleness and almost no ductility (Fig. R10). It is difficult to practically apply without mechanical properties in many service locations, such as space telescopes, LNG ships, and high-voltage transport wire cores. Hence, we try to explore zero thermal expansion alloy with exceptional mechanical response.

Recently, a space gravitational wave detection project was proposed, which needs to achieve high toughness, ZTE performance, and structure forming as the ultra-stable components. Given your suggestions, we have revised our description and added examples to make it more logical: “As application-oriented, ZTE alloys must exhibit not only **a low coefficient of thermal expansion** ($\alpha_l \leq 2.0 \times 10^{-6} K^{-1}$) but also **good toughness to withstand external mechanical loads**. **A typical example is the ultra-stable components in gravitational wave detection (temperature windows ranging from 292.5 - 293.5 K)**”.

Fig. R10 a, The compressive stress-strain curves of typical ZTE intermetallic compounds. **b**, The fracture morphology of ZTE intermetallic compounds.

Comment 2: 182-185 and Fig. 1b: Because of their symmetry, cubic phases have isotropic thermal expansion (independent of whether positive, zero, or negative). Hence, combining two cubic phases results in isotropic thermal expansion, which is also expected for the dual-phase microstructures studied here. However, their thermal expansion is somewhat anisotropic (Fig. 1b). Where do these variations between different sample directions in Fig. 1b come from?

Reply: As you mentioned, due to the cubic symmetry of the two phases, the

alloy should behave isotropic as expected. Therefore, to answer the variations between different sample directions, we conducted additional EPMA measurements (Fig. R11). We observed that there are certain differences in the morphology of the alloy (S-3) between the in-plane (TD-ND) and out-of-plane (LD-TD), the in-plane appearance is a continuous network and the out-of-plane shape is dendritic, which may be related to the growth direction of the as-cast alloy. As a result, the dilatometer thermal expansion behaves with a negligible difference.

In the revised version, we changed the corresponding description to “S-3 displays nearly isotropic ZTE behavior and stable thermal shock resistance (Fig. 1b and Supplementary Fig. 3).” we have added the discussion and corresponding result in the supplementary information.

Fig. R11 The microstructure of the S-3 alloy. **a-g** The electro-probe micro-

analyzer (EPMA) image of the S-3 alloy in-plane (TD-ND) and elements mapping. **h-n** The electro-probe micro-analyzer (EPMA) image of the S-3 alloy out-of-plane (LD-TD) and elements mapping.

Comment 3: *Authors write that “From a lattice viewpoint, there were three crystallographic sites (Fe_I (8b), Fe_{II} (96i), and Fe (2a)) arbitrarily occupied by Fe, Co, and Si atoms”. How do authors reach this conclusion? Via measurements (which?), theoretical calculations (which?). This should be explained.*

Reply: Thanks for your constructive comment. (i) We wrote this description because we found that when we introduce the second phase (α -Fe) into $La(Fe, Co, Si)_{13}$ phase, Co and Si atoms will enter the α phase ($Fe(2a)$ sublattice), and Fe atoms will also enter $La(Fe, Co, Si)_{13}$ phase ($Fe_I(8b)$, $Fe_{II}(96i)$), thus causing the chemical composition of the two phases to change, thereby deteriorating the thermal expansion performance.

(ii) we have carefully checked the occupation of Fe, Co, and Si atoms, respectively. We noticed that in the cubic $NaZn_{13}$ -type structure, Fe and Co atoms occupy two Wycko positions $Fe_I(8b)$ and $Fe_{II}(96i)$, while Si atoms share the $Fe_{II}(96i)$ site with Fe and Co atoms [1, 2]. In the α -Fe phase, all Fe, Co, and Si atoms can occupy the $Fe(2a)$ sublattice, which is confirmed by 3D-APT results.

Thus, we have realized that “From a lattice viewpoint, there were three crystallographic sites (Fe_I (8b), Fe_{II} (96i), and Fe (2a)) arbitrarily occupied by Fe, Co, and Si atoms” is not rigorous. In the revised version, we have changed the description to “From a crystallographic viewpoint, Fe and Co atoms would reside at three distinct crystallographic sites: Fe_I (8b), Fe_{II} (96i), and Fe (2a). Si atoms share the Fe_{II} (96i) and Fe (2a) with Fe and Co atoms. This occupation type results in deteriorating thermal expansion upon introduction of the α phase (Fig. 1d)”. In addition, we revised the crystal structure model in Fig. 1d (Fig. R12).

Fig. R12 The crystal structure of the L and α phase.

- [1] Rosca M., *et al.* Neutron diffraction study of $\text{LaFe}_{11.31}\text{Si}_{1.69}$ and $\text{LaFe}_{11.31}\text{Si}_{1.69}\text{H}_{1.45}$ compounds. *J. Alloy Compd.* **490**, 50-55 (2010).
- [2] Hao J., *et al.* Large enhancement of magnetocaloric and barocaloric effects by hydrostatic pressure in $\text{La}(\text{Fe}_{0.92}\text{Co}_{0.08})_{11.9}\text{Si}_{1.1}$ with a NaZn_{13} -type structure. *Chem. Mater.* **32**, 1807-1818 (2020).

Comment 4: *Fig. 1e and Fig. 2a: The microstructures in these figures look very different, why? The interdendritic L phase visible in Fig. 2a (mostly continuous network) seems to have a different morphology in Fig. 1a (individual islands).*

Reply: We thank the reviewer for bringing this to our attention. We carefully re-examined and analyzed the EBSD and EPMA data (Fig. R13). The microstructures in these figures look very different and may originate from two aspects: **(i)** The huge difference in grain size between the two phases (L phase, $\sim 10\ \mu\text{m}$; α phase, $\sim 200\ \mu\text{m}$). To collect the crystallographic orientation of the α phase by EBSD, we need a larger field of view (scale bar, $200\ \mu\text{m}$, Fig. R13 a, and d), which results in the inability to accurately confirm the L phase of small grains. When collecting data at high magnification (scale bar, $25\ \mu\text{m}$, Fig. R13 b, c, e, and f), the L phase can be accurately confirmed to be a network structure.

(ii) Since there are still some differences between the EBSD at high magnification and the EPMA. To ensure the rationality of the results, we re-synthesized a dual-phase alloy and conducted an EBSD measurement (Fig.

R14). The small-sized L phase still cannot be accurately resolved at low magnification (Fig. R14 a and d), but the two phases show a clear network structure at high magnification (Fig. R14 b, c, e, and f). In this version, we have added the discussion and corresponding result in the revised manuscript and supplementary information.

Fig. R13 a-f, The band contrast and IPFZ images confirmed by EBSD at different magnifications.

Fig. R14 a-f, The band contrast and IPFZ images confirmed by EBSD at different magnifications.

Comment 5: *Fig. 1b: How do LD, TD, and ND relate to the ingot directions?*

Reply: We have added the relationship between ingot direction and coordinate system in the revised supplementary information (Fig. R15).

Fig. R15 Schematic diagram of the relationship between ingot and coordinate system

Comment 6: *I130: Specify what is meant by “natural heterostructures” and “conventional composites”. What are their characteristics?*

Reply: The “natural heterostructures” refer to a dual-phase alloy produced by in-situ synthesis [3], such as the heterostructure structure constructed by the chemical partition in the present work. However, the traditional strategy to attain ZTE composite is powder solid phase sintering, i.e. by sintering two precursor powders [4-6]. These composites usually suffer from undesired microstructures or weak interfacial bonding, resulting in poor overall mechanical properties and thermal cycling performances. More importantly, the ZTE performance is highly composition-sensitive — a slight interfacial mass transfer during high-temperature synthesis may suppress or vanish the ZTE property. In comparison, there was no segregation of deleterious elements (e.g., oxygen, nitrogen) at the interface, which validates the stability of the interface and highlights the

superiority of natural heterostructures over conventional composites.

- [3] Yu C., *et al.* Plastic and low-cost axial zero thermal expansion alloy by a natural dual-phase composite. *Nat. Commun.* 12, 4701 (2021).
- [4] Huang Y., *et al.* Localized magnetic moments variation for strengthening and tuning thermal expansion behavior of Mg alloys. *Acta Mater.* **259**, 119238 (2023).
- [5] Zhou C., *et al.* Near-zero thermal expansion of ZrW₂O₈/Al-Si composites with three-dimensional interpenetrating network structure. *Compos. Part B-Eng.* **211**, 108678 (2021).
- [6] Zhou H., *et al.* Low-melting metal bonded MMX/In composite with largely enhanced mechanical property and anisotropic negative thermal expansion. *Acta Mater.* 229, 117830 (2022).

Comment 7: *Fig. 2g: How were the individual dilatation curves for alpha and L phases obtained? A dilatometer is mentioned in the caption, but how can authors extract expansion curves for individual phases from a dilatometer curve measured on a dual-phase polycrystal? Details must be provided otherwise the analysis remains unreasonable.*

Reply: To obtain the individual dilatation curves for the α and L phases, we first determined the individual compositions based on the results of the 3D-APT. We believe that 3D-APT is very precise in determining the chemical composition of two phases (Fe_{92.42}Co_{4.34}Si_{3.07} (α phase) and LaFe_{10.30}Co_{0.83}Si_{1.87} (L phase)). Second, we synthesized the single-phase Fe_{92.42}Co_{4.34}Si_{3.07} (α phase) and LaFe_{10.30}Co_{0.83}Si_{1.87} (L phase) in a targeted manner, respectively. Hence, we tested their dilatometer thermal expansion, as shown in Fig. R16. The L phase shows negative thermal expansion ($\alpha_l = -37.31 \times 10^{-6} \text{ K}^{-1}$, 260 - 310 K), and the α phase shows positive thermal expansion ($\alpha_\alpha = 10.76 \times 10^{-6} \text{ K}^{-1}$, 260 - 310 K). Based on this, we calculated the dilatometer thermal expansion of the S-3 as the following formulas (1) and (2):

$$\frac{dL}{L_0}(\text{calc.}) = \frac{dL}{L_0}(\alpha) \times \text{Vol.}_\alpha \% + \frac{dL}{L_0}(L) \times \text{Vol.}_L \% \quad (1)$$

$$\alpha_{\text{calc.}} = \frac{dL}{L_0}(\text{calc.})/\Delta T \quad (2)$$

As a result, the coefficient of calculated thermal expansion ($\alpha_{\text{calc.}}$) is $0.91 \times$

10^{-6} K^{-1} (260 - 310 K), which is almost consistent with the dilatometer thermal expansion of S-3 alloy ($\alpha_l = 1.00 \times 10^{-6} \text{ K}^{-1}$, 260 - 310 K). **In the revised manuscript, we added an explanation that the two-phase components are determined through 3D-APT data.**

Fig. R16 The dilatometer thermal expansion of S-3, $\text{LaFe}_{10.30}\text{Co}_{0.83}\text{Si}_{1.87}$ (L phase), $\text{Fe}_{92.42}\text{Co}_{4.34}\text{Si}_{3.07}$ (α phase) alloys. The calculated S-3 ZTE (empty circle point line) is derived from L-phase and α phase thermal expansions.

Comment 8: *I163: "Further microstructural evolution analysis reveals a continuous initiation and accumulation of microcracks in the L phase, which is the predominant mode in this stage (Supplementary Fig. 10)." Check the reference to Supp. Fig. 10, likely Supp. Fig. 11 is meant.*

Reply: Sorry for the mistakes. We have revised it in the manuscript.

Comment 9: *Fig. 3d, e: Why are data only shown in the strain range 0 to 16% even though the uniform elongation is almost 31% (Fig. 3a)?*

Reply: This is based on the following two considerations: (i) We have experience in measuring in-situ load neutron diffraction, so we believe that 15% strain is enough to cover the entire deformation process (both the α and L phase are yielded). (ii) The beamtime of the in-situ load neutron diffraction experiment is very limited, and it takes more than twice the time to 31 % strain. Therefore, we only collected the data at $\varepsilon = 16\%$.

Comment 10: *I181: Explain what is meant by “deep mechanism”.*

Reply: We changed the “deep mechanism” to “forming mechanism” in the revised manuscript: “*Given the forming mechanism for the defect structure, we selected two distinct regions to study their local structure at the atomic level (Fig. 4a).*”

Comment 11: *I195: Authors claim that dislocations in the alpha phase belong to slip system $\{110\} \langle 111 \rangle$, however, do not show slip trace or Burgers vector analysis. Hence, the authors should revise their statement or provide a dislocation analysis.*

Reply: We have deleted the statement of alpha phase belonging to slip system $\{110\} \langle 111 \rangle$ slip system in the revised manuscript.

Comment 12: *I199 See comment about Supp. Fig. 12 d, f.*

Reply: We have added a note to explain that there is some damage to the crack morphology during sample preparation.

Comment 13: *Fig. 5a: Classical Fe-Ni-based Invar alloys should be included in the graph since they are the industry standard and widely used. Linked to that, the authors should briefly discuss the magnetization of the present alloys in relation to Fe-Ni Invar alloys. Can the present alloys be used in magnetic fields?*

Reply: (i) We have conducted the compressive stress-strain measurement of the Invar alloy, which exhibits excellent ductility (Fig. R17 a). Owing to the Invar alloy being a material with good compression ductility, We selected the compressive strength ($\delta = 0.68$ GPa) at $\varepsilon = 30\%$ to compare in Fig. 5b (Fig. R17 b).

(ii) We have supplemented the magnetic curve of Invar alloy under 100 Oe (Fig. R18). Both the S-3 alloy and the Fe-Ni Invar alloy ferromagnetic order. As

we know, the magnetization of ferromagnetic order material may be affected by the magnetic fields, so the thermal expansion performances may also have some changes under the magnetic fields. **We have added the data and discussion of Invar alloy in the revised manuscript.**

Fig. R17 a, The compressive stress-strain measurement of the Invar alloy. **b**, Mechanical properties of ZTE alloys. The comparison of the compressibility at fracture (ϵ_f) and ultimate strength (δ_{us}) of various ZTE alloys. The hollow and solid dots refer to anisotropic and isotropic thermal expansion, respectively. It is noted that Invar is a completely plastic material, for comparison, we used the compressive strength at 30 % strains here.

Fig. R18 Field-cooling (FC) magnetization (M) of Invar alloy at a magnetic field (100 Oe).

Comment 14: Fig. 5b: Authors should explain the fabrication process of the rods/tubes shown.

Reply: We have added the fabrication process of the rods/tubes in methods:

“The rods/tubes are melted under induction melting conditions and cast into Φ 25 mm diameter rods, annealed under vacuum (1373 K + 24 hours), then machined to suitable size and surface polished” .

Comment 15: Explain the “Counts” axis in Supp. Fig. 5. What does it represent?

Reply: It refers to the number of diffraction spectra collected at different angles. As shown in Fig. R19, the sample can rotate from 45° to 90° inside the LD-TD plane (Ω) and rotate in the TD-ND plane (θ) from 0 to 360° . The 75 counts are used to collect the three-dimensional crystallographic information (11.25 degrees/step for 45-degree vertical rotation; 24 degrees/step for 336-degree horizontal rotation). At each step, the two detectors will be collected, respectively.

We have added the explanation in the supplementary information in the revised manuscript.

Fig. R19 Schematic diagram of the in-situ neutron diffraction experimental set-up from the top view. The sample is horizontal and positioned at 45° from the incident beam such that Bank 1 probes the strain component along the LD, while Bank 2 simultaneously probes the strain component in the TD, as shown in two insets.

Comment 16 I289: Authors write that dL/L_0 is the dimensional change per unit temperature. If so, then there would be no need to divide dL/L_0 by Delta T in

(3). More likely, dL/L_0 is simply the relative length change across the temperature interval ΔT .

Reply: You are right, what we expressed was misunderstood. dL/L_0 is simply the relative length change across the temperature interval ΔT . In the version, we revised it as: “Here, $\frac{dL}{L_0}$ is the dimensional change across the temperature interval ΔT . Vol. is the volume percentage. α_l represents the coefficient of linear expansion.”

Comment 17 I291: *How was strain measured?*

Reply: Sorry for the mistake. We changed it to “The compressive stress-strain curves were measured using a CMT4105 universal electronic machine at room temperature, with a Φ 5×10 mm cylinder and an initial strain rate of $1.0 \times 10^{-3} \text{ s}^{-1}$. Each sample was tested a total of four times.”

Comment 18 I301: *Eq. (4) misses an equal sign.*

Reply: We have added the equal sign in the revised manuscript.

Comment 19 Supp. Fig. 12 a,b,c: *Authors should provide clean and sharp images of the stacking faults typically showing fringes when viewed at an inclination angle (for instance Fig. 2 in <https://normandie-univ.hal.science/hal-02175446>).*

Reply: Thanks for your insightful suggestions and comments. (i) We have read this reference carefully and added it as a reference in the revised manuscript. In addition, we have re-conducted the TEM experiment to observe the stacking fault structure (Fig. R20). Three independent areas were selected, and we tried different rotation angles and multiples to obtain the best stacking fault morphology, but we still got this single linear result, which we believe is closely related to the crystal structure of the sample.

To confirm that our results are credible, we have selected another BCC alloy with stacking fault deformation for TEM experiments. This result is very

similar to the reference (Fig. R21). Therefore, we believe that the observed results in the highly ordered complex $\text{La}(\text{Fe}, \text{Co}, \text{Si})_{13}$ phase are intrinsic.

Fig. R20 Three dependent stacking fault structures observed at different magnifications.

Fig. R21 Stacking faults structure in BCC alloy at $\varepsilon = 5\%$.

Comment 20 Supp. Fig. 12 d, f: Microcracks in f seem to be holes originating from the TEM lamella thinning process. Also, for two “microcracks” suggested by yellow dashes in d and f no cracks are seen in the images.

Reply: (i) We have considered that the holes are affected by the TEM lamella thinning process in Supplementary Fig. 12 d and f. Therefore, we tested SEM at 15% strain, and some shear cracks were indeed observed in the L phase (Fig. R22 a). At the same time, during the TEM test, we noticed that the number of cracks appearing at $\varepsilon = 15\%$ strain was significantly greater than $\varepsilon = 2\%$ and 5%, so we believe that the cracks are caused by intrinsic deformation. (ii) Sorry for the misleading of the two “microcracks” suggested by yellow dashes in d and f, it is a stacking fault (Fig. R22 b). **In the revised manuscript, we have added the description that the cracks were affected by the thinning process. We have revised the misleading of the two “microcracks” suggested by yellow dashes in d and f to stacking faults.**

Fig. R22 a, The microstructure of the S-3 alloy at $\epsilon = 15\%$ strain observed by SEM. **b**, The microstructure of the S-3 alloy at $\epsilon = 15\%$ strain determined by TEM.

Comment 21 *The sentence in I34 misses a verb.*

Reply: We have added a verb in the revised manuscript: “Zero thermal expansion (ZTE) in alloys, **is** generally manipulated by rigorous spin-lattice-orbital coupling, emerging in a variety of intermetallic compounds.”

Comment 22 *I36: slip systems of independence ==> independent slip systems*

Reply: We have changed it to “*independent slip systems*”.

Comment 23 *I239: Authors write: “...arc melting with the element more than 99.9% under high purity argon...” Do they mean “...arc melting of elements with purity better than 99.9% under high purity argon...”?*

Reply: Sorry for the misleading. We have revised it to “The series of $\text{LaFe}_{0.939x}\text{Co}_{0.061x}\text{Si}_{0.0583x}$ ($x = 37.5, 47.5, 57.5,$ and 67.5 , labeled as S-1, S-2, S-3, and S-4, respectively) were prepared by arc melting of elements with purity better than 99.9% under a high-purity argon atmosphere.”

Comment 24 *I281: Delete the single “a” in thermal expansion a.*

Reply: We have deleted the “a”.

Comment 25 *The sentence starting on l291 is incomplete.*

Reply: We revised the sentence as “*The compressive stress-strain curves were measured using a CMT4105 universal electronic machine at room temperature, with a Φ 5×10 mm cylinder and an initial strain rate of 1.0×10^{-3} s⁻¹. Each sample was tested a total of four times*”.

Comment 26 *Exploring zero thermal expansion (ZTE) alloys with exceptional mechanical response is crucial for their practical use. □ Zero thermal expansion (ZTE) alloys with exceptional mechanical response are crucial for practical applications.*

Reply: We have changed it to “*Zero thermal expansion (ZTE) alloys with exceptional mechanical response are crucial for practical applications.*” in the revised manuscript.

REVIEWER COMMENTS

Reviewer #1 (Remarks to the Author):

In the revised manuscript, the authors have well addressed my previous questions. From the point of view of the importance, innovation and technicality, this work has met the publication requirements.

Reviewer #2 (Remarks to the Author):

I have examined the revised manuscript. The authors have clarified the unclear points. It is acceptable to me now.

Reviewer #3 (Remarks to the Author):

In the introduction the authors write: "Such compounds tend to be brittle because of their [...] independent slip systems". I believe what the authors mean is that they are brittle because of the **lack of** independent slip systems. Independent slip systems facilitate ductility after all. The authors should explain their reasoning and, where appropriate, modify that formulation accordingly.

Fig. 2g: The authors added the information to the manuscript that the two-phase components were determined through 3D-APT. To make the analysis approach transparent, it should be also mentioned in the manuscript in the 'Materials and Preparations' paragraph that single-phase $\text{Fe}_{92.42}\text{Co}_{4.34}\text{Si}_{3.07}$ (α phase) and $\text{LaFe}_{10.30}\text{Co}_{0.83}\text{Si}_{1.87}$ (L phase) were synthesized (as the authors explained to the reviewer in their responses).

Reviewer #1 (Remarks to the Author):

In the revised manuscript, the authors have well addressed my previous questions. From the point of view of the importance, innovation and technicality, this work has met the publication requirements.

Reply: We are very grateful to the reviewers for their approvement of the current work and for their considerable time and effort.

Reviewer #2 (Remarks to the Author):

I have examined the revised manuscript. The authors have clarified the unclear points. It is acceptable to me now.

Reply: We appreciate the referee's positive feedback and the time and effort they put into this work.

Reviewer #3 (Remarks to the Author):

Comment 1: *In the introduction the authors write: "Such compounds tend to be brittle because of their [...] independent slip systems". I believe what the authors mean is that they are brittle because of the lack of independent slip systems. Independent slip systems facilitate ductility after all. The authors should explain their reasoning and, where appropriate, modify that formulation accordingly.*

Reply: Thank you for pointing out the problem carefully. The reason for this problem may be that we mistakenly deleted "insufficient" in the manuscript. In the revised version, we have changed it as "*Such compounds tend to be brittle because of their multiple covalent or ionic bonds and the lack of independent slip systems.*"

Comment 2: *Fig. 2g: The authors added the information to the manuscript that the two-phase components were determined through 3D-APT. To make the analysis approach transparent, it should be also mentioned in the manuscript in the 'Materials and Preparations' paragraph that single-phase $\text{Fe}_{92.42}\text{Co}_{4.34}\text{Si}_{3.07}$ (α phase) and $\text{LaFe}_{10.30}\text{Co}_{0.83}\text{Si}_{1.87}$ (L phase) were synthesized (as the authors explained to the reviewer in their responses).*

Reply: In the revised version, we have added the description in the Materials and Preparations, Methods: "*Besides, to obtain the individual dilatation curves for the α and L phases, we first determined the compositions of two phases ($\text{Fe}_{92.42}\text{Co}_{4.34}\text{Si}_{3.07}$ for the α phase and $\text{LaFe}_{10.30}\text{Co}_{0.83}\text{Si}_{1.87}$ for the L phase) based on the results of the 3D-APT. Second, we synthesized the single-phase $\text{Fe}_{92.42}\text{Co}_{4.34}\text{Si}_{3.07}$ (α phase) and $\text{LaFe}_{10.30}\text{Co}_{0.83}\text{Si}_{1.87}$ (L phase) in a targeted manner, respectively.*"

REVIEWERS' COMMENTS

Reviewer #3 (Remarks to the Author):

Authors have properly responded to the last remarks. No further comments.